**Cite this article:** Çakır-Mete B, Ergün AF and Şafak A (2026). Mental health needs, stressors and coping resources of internally displaced children in post-conflict Syria: A qualitative study with NGO staff. *Cambridge Prisms: Global Mental Health*, **13**, e25, 1–9

post-conflict; internally displaced children; mental health needs; Syria; qualitative research

**Corresponding author:**
Betül Çakır-Mete;
Email: betul.mete@stu.ihu.edu.tr

# Mental health needs, stressors and coping resources of internally displaced children in post-conflict Syria: A qualitative study with NGO staff

Betül Çakır-Mete , Ahmet Faruk Ergün and Ayşe Şafak

Ibn Haldun University, Türkiye

## Abstract

Although the needs of conflict-affected children are well-documented, research on the post-conflict period is limited, particularly in Syria, where the fall of the Assad regime has created a rapidly evolving environment for internally displaced children. This study explores how key informants perceive the mental health needs, daily stressors and coping strategies of internally displaced children during the post-regime period. Online semi-structured interviews were conducted with 10 staff members from a non-governmental organization working in psycho-social support in the Syria camps. Data were analyzed using thematic analysis. Five major themes emerged: (1) stressors in the current camp environment, (2) challenges related to return, (3) observed emotional and behavioral difficulties, (4) children's psychological resources and (5) needs and gaps in support services. Findings highlight the inseparability of children's mental health from basic needs, the role of place-based attachments in return processes and the importance of a holistic approach that considers context-specific stressors and resources in this unique period.

## Impact statement

This study investigates the mental health needs of internally displaced children living in the Atmeh camp in northern Syria during the post-conflict period following the fall of the Assad regime. Using qualitative insights from non-governmental organization (NGO) workers who work closely with these children, the study sheds light on an often-overlooked issue in global mental health: the psychological challenges that emerge not only during displacement but also throughout the return and reconstruction phases. Despite a prevailing optimism that post-conflict recovery will naturally lead to improved wellbeing, the findings suggest that return processes can be equally complex and emotionally demanding. The study highlights that the mental health of internally displaced children in post-conflict Syria is deeply intertwined with their basic needs, place-based attachments and culturally and religiously informed psychological resources. The results emphasize the need for context-specific and holistic psychological interventions that address both material and cultural aspects of wellbeing. Mental health initiatives designed for post-conflict contexts should be sensitive to the lived experiences and value systems of affected children. For policymakers and humanitarian organizations, investing in such programs is essential to support healthier adaptation and to empower the generation that will contribute to rebuilding post-conflict Syria.

## Background

Current estimates indicate that conflict affects approximately one in eight people globally and one in six children, with around 120 million people forcibly displaced (ACLED, 2024; UNICEF, 2024; UNHCR, 2025a). Throughout the prolonged conflict preceding the fall of the Assad regime in December 2024 (Wedeen, 2025), Syria emerged as the global leader in internally displaced populations (IDPs), with children and women constituting 80% (IDMC, 2025). Many internally displaced children continue to reside in camps across Syria including Atmeh, which hosts approximately two million Syrian IDPs last year (Anadolu Agency, 2025). Within these settings, IDPs are exposed to overlapping risk factors for poor mental health, including chronic overcrowding, economic deprivation, resource scarcity, environmental instability and lack of essential services (Owoaje et al., 2016). Consistent evidence from displaced populations indicates high prevalence rates of post-traumatic stress disorder (PTSD), depression and anxiety, with children disproportionately affected (Morina et al., 2018; Blackmore et al., 2020). Studies focusing on Syrian war-affected children report similar patterns, documenting extensive exposure to

traumatic and chronic stressors and a pooled PTSD prevalence of 36% based on a meta-analysis of 26 studies (Oleimat et al., 2023; Kanan and Leão, 2024).

Existing literature has predominantly examined the mental health needs within refugee children resettled in host countries, and their findings are explicitly noted as not fully generalizable to internally displaced children (Kampalath et al., 2023). An anthropological field study comparing Syrian refugees in Europe with Syrian IDPs living in camps demonstrated that internally displaced children experience more severe deprivation, including unmet basic needs, economic precarity, child labor and restricted mobility, leaving them largely confined to camps and focused on survival (Wessels, 2018). Evidence from both a case study conducted in Atmeh Camp and a scoping review examining children living in Syrian IDP camps consistently indicates that health needs in IDP camps exceed available capacity, with children facing elevated risks of inadequate vaccination, malnutrition, anemia, and physical abuse, while mental health needs remain underreported due to structural barriers such as the lack of private consultation spaces (Aburas et al., 2018; Kampalath et al., 2023). Consistent with these findings, a study of children aged 4–10 years living in IDP camps in northern Syria indicated higher levels of emotional symptoms than those reported among refugee children resettled in Western countries (Cartwright et al., 2015).

In contrast to the literature focusing on refugee children, studies that examine children living directly in IDP camps in Syria have faced substantial structural constraints, including prevailing insecurity, restricted physical access and funding shortfall (Quosh et al., 2013; Kampalath et al., 2023; Harphoush et al., 2025). Existing studies – primarily using surveys or interviews with caregivers – document elevated emotional and behavioral difficulties among children including nightmares, enuresis, aggression, withdrawal and fearfulness (Cartwright et al., 2015; El-Khani et al., 2018; Baroudi and Humeydi, 2024; Hatahet et al., 2024), with approximately one in five children reported to have been exposed to violence (Vernier et al., 2019). Consistent with these findings, evidence from different regions in Syria indicates that internally displaced children bear a higher burden of mental and physical health problems – including elevated post-traumatic stress disorder (PTSD) prevalence – and show greater reliance on NGO-run pediatric services than their non-displaced peers, reflecting prolonged stress, cumulative trauma exposure and chronically unstable living conditions (Perkins et al., 2018; Raslan et al., 2021; Kampalath et al., 2024).

Following the fall of the Assad regime in 2024, active conflict in Syria has substantially decreased, and emerging data indicate an estimated 1,955,879 IDPs returned within Syria, primarily concentrated in the Aleppo, Idlib, Hama and Homs governorates, between 27 November 2024 and 11 December 2025, including over one million departures from IDP sites (UNHCR, 2025a,b,c). Although the post-conflict period is often framed as a transition toward stability, evidence from conflict-affected settings suggests that mental health needs frequently persist or even intensify despite gradual improvements in basic living conditions, due to ongoing structural and psychosocial stressors such as bereavement, family disruption, economic insecurity and prolonged service gaps (Crombach et al., 2017). In post-conflict contexts, difficulties related to meaning-making, self-blame, persistent rumination and identity disruption have also been widely documented, further complicating psychological recovery (Farwell, 2003; Poudyal et al., 2009; Zúñiga and Hamann, 2015). Consistent with these post-conflict vulnerabilities, a recent scoping review identified child mental health as the

most frequently documented concern, and reported that nearly 50% of children in government-controlled areas exhibit PTSD symptoms, alongside comparably high prevalence rates in non-government-controlled regions in post-conflict Syria (Harphoush et al., 2025).

In light of these considerations, despite growing attention to the mental health of war-affected children, research has largely focused on refugee populations, while the distinct needs of internally displaced children living in camps – particularly in Syria – have received comparatively less attention. Although some qualitative studies in Syrian camps have engaged NGO workers, educators, doctors or local officials as key informants, these studies have largely focused on acute crisis contexts, often positioning informants as data sources rather than as holders of contextual expertise, and have shown limited methodological rigor (Quosh et al., 2013; El-Khani et al., 2018; Wessels, 2018; Oleimat et al., 2023). Moreover, the post-conflict phase remains underexplored due to the recency of this historical transition; although a limited number of studies have examined return processes, these have focused on earlier phases prior to the fall of the regime and have not centered on children's mental health needs (e.g. Solymári et al., 2025). Against this background, children's mental health in IDP camps during the post-conflict period warrants focused attention, as it theoretically addresses the gaps in the literature and practically informs mental health interventions for children facing additional post-conflict pressures. Positioned within this gap, the present study draws on the perspectives of key informants working in Atmeh Camp, Syria, to address the following research question: How do key informants perceive the post-conflict mental health needs, daily stressors and coping mechanisms of internally displaced children living in Atmeh Camp, Syria, in the aftermath of the fall of the Assad regime?

## Methods

### Participants

Participants were recruited using purposive criterion sampling to obtain informed perspectives from professionals working closely with internally displaced children in Atmeh Camp, near the Syrian–Turkish border, which hosts a large displaced population. Given the high-risk and volatile context, direct engagement with children was avoided in line with the *do no harm* principle, as intensive research inquiries may risk re-traumatization and children's self-reports in crisis settings can be constrained by immediate survival needs and developmental limitations. Accordingly, the study relied on key informants with sustained professional involvement in children's care.

Inclusion criteria required participants to be Syrian professionals, ensuring an insider perspective, with direct or indirect experience working with children in Atmeh Camp for at least one service term (approximately 5–6 months), and to hold an active role within an NGO formally engaged with the camp. Exclusion criteria included NGO staff without camp-related responsibilities, individuals from different cultural or ethnic backgrounds, and those with limited tenure (≤2–3 months), due to insufficient contextual exposure. Participant demographics are presented in Table 1.

Recruitment followed a gatekeeper-mediated process through an NGO operating in Atmeh Camp. After the study aims – examining children's mental health needs, daily stressors and coping mechanisms, with the dual purpose of generating academic and practice-oriented insights – were communicated to potential

**Table 1.** Demographic characteristics of the participants

|  | Gender | Age | Familiarity with the camp |
|---|---|---|---|
| P1 | Female | 46 | 7 years indirect experience |
| P2 | Female | 33 | 12 years direct experience |
| P3 | Female | 50 | 5 years direct experience |
| P4 | Female | 45 | 5 years direct experience |
| P5 | Female | 28 | 2 years indirect experience |
| P6 | Female | 22 | 5 months indirect experience |
| P7 | Male | 56 | 5 years direct experience |
| P8 | Male | 36 | 6 years direct experience |
| P9 | Male | 40 | 5 years direct experience |
| P10 | Male | 52 | 11 years direct experience |

*Note*: Direct experience refers to participants who have lived inside the camp. Indirect experience refers to those who have not lived in the camp but have worked in the camp area.

participants via the NGO field coordinator, eligible key informants were identified. Participants received an information sheet, provided written informed consent, explicitly consented to audio recording, and were informed that participation was voluntary and withdrawal possible at any time without consequences. No incentives were offered. Interviews were conducted online via Google Meet at times that did not interfere with humanitarian duties.

### Procedure

Semi-structured qualitative interviews were conducted in May 2025, with an average duration of 40 min; the interview guide is provided in the Appendix. Most interviews were supported by a native Arabic-speaking interpreter trained in psychological counseling and qualitative interview techniques, while two Turkish-speaking participants were interviewed directly. Participants were informed about the interpreter's role, who signed a confidentiality agreement and had no involvement in participants' professional networks. Interviews were conducted by the first two authors.

### Analysis

Data analysis followed an iterative thematic analysis involving three researchers, guided by Creswell (2012). Audio recordings were transcribed by the first two authors, who also conducted the interviews, ensuring close familiarity with the data. All researchers repeatedly read the transcripts prior to and during analysis. Line-by-line coding was conducted using a hybrid deductive–inductive approach, with predefined interview topics guiding initial codes while allowing new themes to emerge inductively. Codes were iteratively organized into categories and overarching themes and systematically cross-checked through regular meetings between the first two authors. Data collection concluded after 10 interviews when thematic saturation – defined by the recurrence of identical codes – was reached.

The third researcher reviewed all transcripts and the coding hierarchy as a peer debriefer, contributing to theme refinement. Credibility was ensured through investigator triangulation and peer debriefing; confirmability and transparency through the use of a semi-structured interview guide as an audit trail and thick descriptions supported by direct quotations; and dependability through systematic coding and cross-checking procedures. Finally, all

researchers returned to the original transcripts to ensure accurate representation of participants' accounts. All data were securely stored and will be destroyed upon study completion. AI-assisted tools were used solely for linguistic support during English translation and refinement.

## Results and discussion

Thematic analysis yielded five overarching themes: (1) stressors in the current camp environment, (2) challenges related to return, (3) observed behavioral and emotional difficulties, (4) children's psychological resources and (5) needs and gaps in support services. The themes and their corresponding categories are summarized in Table 2.

### Stressors in the current camp environment

Participants' accounts of the stressors present in the camp environment primarily clustered around two categories: (1) economic hardship and (2) the lack of safe spaces. A distinctive finding was that, despite the formal end of the conflict, most families remain unable to return to their places of origin and continue to experience significant deprivation within the camp. These stressors closely mirror those reported in the IDP literature, particularly unmet basic needs that pose risks to both physical and mental health (Allen et al., 2014; Owoaje et al., 2016; Pourmotabbed et al., 2020). Economic hardship compelled many children to engage in labor, while families remained highly dependent on humanitarian aid; interruptions in aid delivery were described as acute threats to survival, often resulting in shortages of food and water. Participants further noted that the gradual reduction of humanitarian assistance in the post-conflict period may intensify these vulnerabilities. In parallel, the lack of safe spaces was consistently highlighted, with children exposed to swearing, bullying and maladaptive behaviors through unsupervised socialization within the camp.

The current study further illustrates how these challenges affect children both directly and indirectly through their families. Participant 2 highlighted the severity of economic hardship and its link to persistent feelings of loss: "The children say: 'We have already been displaced; we lost our belongings. Then we moved into a small tent, and once again, due to the water and rain, we lost everything'." Similarly, witnessing their parents' ongoing struggles, children

**Table 2.** Themes and categories

| Themes | Categories |
|---|---|
| Stressors in the current camp environment | Economic hardship, lack of safe spaces |
| Challenges related to return | Adverse circumstances in return areas; identity confusion and adjustment problems; uncertainties about future |
| Observed behavioral and emotional difficulties | Behavioral difficulties; stress and anxiety difficulties; trauma-related difficulties |
| Children's psychological resources | Protective factors, coping mechanisms |
| Needs and gaps in support services | Target groups required psychological support; the nature of psychological and social support needs; the importance of a culturally and religiously sensitive approach |

frequently internalized the stress and, as Participant 9 noted, sometimes experienced guilt, believing that their presence worsened their family's circumstances: "Children express such thoughts of themselves: 'If it weren't for us, our family's situation wouldn't be like this. It could be better'."

## Challenges related to return

Based on participant accounts, the challenges related to the return process can be grouped into three main categories: (1) adverse circumstances in return areas, (2) identity confusion and adjustment problems and (3) uncertainties about the future. Participants described return areas as marked by persistent insecurity and severe infrastructural damage. Bronfenbrenner's ecological systems theory (2005) and evidence highlighting the importance of safety and stable routines for children's mental well-being (Shonkoff et al., 2011; Selman and Dilworth-Bart, 2023), underscore the centrality of stability; however, participants' accounts consistently pointed to its absence within newly re-established systems. They reported incidents of theft, assaults, killings and the continued presence of individuals affiliated with the former regime, heightening children's fears of renewed violence. Many villages, homes and schools remain destroyed, with reconstruction expected to take years; some families reportedly returned by pitching tents beside ruined homes, with participants citing fatal and injurious building collapses. These accounts underscore that children's mental health cannot be meaningfully addressed without considering ongoing structural vulnerabilities, including insecurity and infrastructural collapse, which persist into the post-conflict period and continue to undermine psychological recovery.

Beyond material and security concerns, the impact of returning on identity confusion and adjustment problems emerged as another key category. For many children, the return does not signify "going home" but rather entering an unfamiliar and unsettling environment. Having spent most or all of their lives in tent camps, many children are too young to remember their original homes. As a result, the places they are expected to reconnect with often feel alien or emotionally distant. The literature highlights that children's place-based attachments form primarily through daily experiences rather than abstract sociocultural meanings, unlike in adults (Twigger-Ross and Uzzell, 1996; Hay, 1998; Morgan, 2009). Consequently, Atmeh Camp may hold more emotional significance for these children than their hometowns, making the return process emotionally distressing. Furthermore, participants noted that reunions with extended family can evoke discomfort rather than comfort, as they may entail social expectations that children find overwhelming.

Adapting to new social environments also presents significant challenges. According to the participants accounts, children raised in the relatively homogeneous camp context – where schools strictly segregated genders and most residents shared similar religious backgrounds – often struggle to adjust to the social diversity and mixed-gender settings in their villages. Such shifts may evoke feelings of alienation and intensify identity-related conflicts. Participant 5 illustrated this sentiment with the statement of a young Syrian who said, "I feel like a stranger in my homeland; I don't belong here or there," reflecting a profound sense of dislocation. The broader literature on return migration has similarly described identity confusion as one of the key difficulties among returnees (Zúñiga and Hamann, 2015). However, for children who have spent much of their lives in camps, these challenges are likely deepened by factors such as age at displacement, duration of camp life and the contrasting social norms between camp and village environments.

Uncertainty emerged as another major source of distress affecting both children who returned and those still residing in camps. Participants described parental indecisiveness – whether to remain in the camp or resettle in the village – as a persistent source of anxiety. Children often expressed distress about not knowing where they would live, with whom, or whether they would have access to education. As noted in prior studies, unstable contexts heighten uncertainty, which in turn contributes to psychological strain (Anderson et al., 2024). Some participants even reported that children perceived the uncertainty of the post-conflict period as more distressing than the hardships experienced under the previous regime. Participant 6 illustrated this through a child's statement: "I wish the regime hadn't fallen; then we wouldn't have to go through all this."

Additionally, although families considered the return to be voluntary, children's emotional responses suggest that, for some, it may have felt like a forced return driven by their parents' decisions. While existing studies often associate psychological distress with forced displacement (Kienzler et al., 2018; Hazer and Gredebäck, 2023), these findings suggest that even "so-called voluntary" returns can evoke similar challenges for children. This highlights the importance of incorporating children's perspectives into return planning, including gradual transitions and participatory approaches (Vathi and Duci, 2015).

## Observed behavioral and emotional difficulties in children

Participants described a wide range of emotional and behavioral responses observed in children primarily emerging in relation to the post-conflict period. These observations were grouped into three main categories: (1) behavioral difficulties, (2) stress- and anxiety-related difficulties and (3) trauma-related difficulties. Participants reported a range of behavioral issues observed among children in the camp, including irritability, reluctance, inappropriate language, oppositional attitudes, aggression, bullying, lying, stealing and other disruptive behaviors. These accounts align with prior research linking disrupted place attachment to behavioral dysregulation (Bunn et al., 2023; Rodriguez-Perez and Castellanos, 2024). According to the participant, who is a psychological counselor, many of these behaviors may reflect unmet emotional needs – particularly the need for attention and difficulties in regulating anger. Thus, beyond inclusion in decision-making processes, providing children with opportunities to express emotions – especially anger – may be helpful for psychological adjustment.

Nearly all participants reported observing signs of psychological distress in children including sadness, grief, disillusionment and anxiety. For those remaining in camps, the departure of teachers and peers disrupted social ties, leading to loneliness and loss of motivation. Participant 4 described how children began skipping exams and classes, saying, "We'll go back anyway, so I don't need to study," illustrating a sense of hopelessness resembling learned helplessness. Similarly, Participant 3 noted that uncertainty about the future evokes anxiety reminiscent of the early displacement period. These experiences are compounded by concrete fears – such as re-displacement and safety. Participant 1 illustrated this vividly: "The children are worried about being forced to migrate again. They are also worried about returning to their villages because there are still cases of murder and theft there."

Participants also noted that trauma symptoms frequently re-emerged during the return phase. For children who had

experienced conflict-related trauma, exposure to familiar places, sensory cues, or discussions about return often triggered painful memories. In group settings, children became visibly distressed when reminded of deceased relatives or the destruction of their homes, indicating that conversations about "return" can serve as powerful emotional triggers. Participant 6 recalled: "Near the school, the sound of gunfire came during a training; some children dropped to the ground, while others looked to the sky for bombs." Such immediate reactions exemplify trauma responses that remain easily reactivated in insecure environments. Beyond direct trauma, participants highlighted children's exposure to secondary trauma – for instance, through family discussions or distressing social media content about wartime atrocities. Participant 8 described young children's anxiety when their fathers left the camp: "Even 4- or 5-year-olds are so aware that they try to stop their fathers from leaving Atmeh, fearing fighter jets or the old regime might attack them."

Anxiety and trauma-related symptoms are well-documented in contexts of war and displacement (Morina et al., 2018). However, since this study was conducted in the early post-conflict period, it is crucial to interpret these symptoms in light of ongoing insecurity. If the broader context is ignored, children's hyperarousal, vigilance or anxiety may be mistaken for psychopathology rather than adaptive survival responses to persistent threats (van der Kolk, 2014). Understanding trauma as a continuing experience rather than a past event thus offers a more accurate and compassionate framework for addressing children's psychological needs.

### Children's psychological resources

Participants reported that, despite a range of behavioral and emotional difficulties, many children possess psychological resources that help them navigate adversity. They described these resources in two interrelated dimensions: (1) protective factors and (2) coping strategies. Protective factors identified by participants include parental support, access to education and engagement in activities promoting emotional regulation. Consistent with previous research highlighting the buffering effects of social support in conflict-affected populations (Tol et al., 2013; Huynh and Li, 2024), participants emphasized that psychologically stable and supportive caregivers help children navigate difficult conditions and improve their mental health. Small gestures – like playing games, doing activities together or accompanying children to school – were viewed as significant contributors to children's well-being. Several participants noted that the benefits of family care are often visible in children's demeanor: those who receive consistent support appear cleaner, more confident and socially engaged than peers who lack such care.

Access to education – even under limited conditions – was also described as a key protective factor. Supported by NGOs, children can regularly attend school, which contributes not only to academic development but also to social connectedness, fostering routine, belonging and future orientation. Schools and community organizations offering vocational training were seen as helping children develop skills and envision future pathways. Participants noted that incorporating emotion regulation activities into educational settings enhances children's coping with stress. Designated activity rooms allow children to engage in games, drawing and other expressive activities, which they enjoy for extended periods. Participant 6 observed that younger children (7–10) are especially drawn to play-based activities, while older children (11–16) prefer art-based tasks, highlighting the importance of developmentally

appropriate options for emotional expression. According to Participant 2, children often show more positive attitudes for up to a week following such activities, suggesting noticeable improvements in emotional regulation skills. These practical observations are consistent with recent research demonstrating that self-narrative art therapy can significantly reduce PTSD symptoms among war-affected Syrian children (Kalthom et al., 2025). Such evidence reinforces the therapeutic potential of integrating expressive art activities into educational settings as a supportive resource for emotional recovery.

Children also employ individual coping strategies, with religious meaning-making frequently emphasized by participants. Most families in Atmeh Camp are religiously observant, and children in this religious context often use faith-based interpretations – such as seeing suffering as a test from God – to make sense of difficulties and develop resilience. Learning about the lives of religious figures, such as the Prophet and his companions, provides positive role models and reinforces values like patience, hope and perseverance. Religious rituals, particularly prayer, were also noted as sources of emotional relief and psychological comfort. These observations highlight the importance of providing religion-sensitive support that aligns with children's faith practices. Furthermore, while religion can sometimes be exploited to fuel social divisions or violence (Appleby, 2000), these observations suggest that religious frameworks could be intentionally utilized to promote values such as compassion, empathy and patience. Especially in the context of the anticipated "return movement" – where children may interact with diverse social groups – leveraging these prosocial teachings can play a vital role in fostering social cohesion and peace-building.

Another coping strategy identified by participants is future-oriented thinking. Despite significant stressors such as destruction, displacement and uncertainty – particularly in the post-conflict period – children were also described as hopeful and optimistic. While returning can disrupt familiar routines, it appears to rekindle aspirations and foster a stronger sense of purpose. Participants observed that many children show enthusiasm about returning to their villages and increased motivation at school. Some express their future goals – such as desired professions – during school events, reflecting a forward-looking mindset. According to Participant 1, this orientation toward rebuilding and contributing to their country illustrates how a sense of national belonging and responsibility can serve as a source of both individual and collective recovery: "Some children have started to look to the future with more hope, saying things like, 'Let's study, let's get a job…we will rebuild our village, reopen our schools'."

Children's reliance on religious meaning-making and future-oriented thinking echoes findings from prior studies (Chow et al., 2020; Serrano et al., 2021; Huynh and Li, 2024). These coping mechanisms not only facilitate adaptation in the face of adversity but may also foster growth, as suggested by post-traumatic growth (PTG) theory. Specifically, religious meaning-making corresponds to the spiritual change domain of PTG, while future-oriented thinking – by motivating children to envision and work toward a better future – may be linked to the domain of new possibilities (Tedeschi and Calhoun, 1996). Furthermore, interpreting adversity as a religious test, acknowledging its temporary nature or seeking opportunity in uncertainty can function as cognitively adaptive strategies that support resilience and agency over passivity (Pargament, 1997; Parsons et al., 2016). Additionally, drawing on collective and religious values – especially when accompanied by a sense of responsibility and belonging – can enhance not only individual coping but also community-level resilience (Kirmayer

et al., 2009; Frounfelker et al., 2020). Taken together, the presence of protective factors and coping strategies during periods of hardship may serve preventive and therapeutic functions and also promote long-term psychological well-being in children (Werner, 2012; Veronese et al., 2019).

### Needs and gaps in support services

Participants discussed the limited availability of services and resources addressing the mental health needs of internally displaced children in Atmeh Camp. The accounts revealed three categories: (1) target groups requiring psychological support, (2) the nature of psychological and social support needs and (3) the importance of a culturally and religiously sensitive approach. Children who have returned to their original homes, those still in camps, and those raised in areas formerly under regime control were all identified as needing context-specific support. Returning children face insecurity, poor infrastructure and adjustment difficulties while children remaining in camps struggle with harsh living conditions, uncertainty and separation. Those resettling in formerly regime-controlled areas may carry the mental health effects of prolonged authoritarian control and systemic hardship. These observations highlight that psychological support needs vary across these groups, requiring contextually informed interventions. Participant 5 reflected on this evolving understanding: "We thought Syria was just Idlib because we were stuck there for two years. After being liberated, we realized the country is much bigger and the need much greater."

Participants also emphasized the need for both psychological and social support delivered by trained professionals and integrated across children's immediate environments – families, schools and communities, consistent with existing research (Pop et al., 2025). Participant 10 described this as a three-tiered approach, where psychoeducation is provided first to the family, then to the school, and finally to the child, ensuring support at each level.

Another key category was the importance of cultural and religious sensitivity in addressing children's psychological needs. Integrating familiar religious teachings – such as stories from the Qur'an or the Prophet's life – was described as an effective way to help children relate to therapeutic content and feel safe during interventions. They noted that respecting cultural and religious values enhances children's receptivity to interventions. As Participant 6 noted: "Sometimes, when we approach a child only from an academic perspective, it feels unfamiliar. But when we approach them with religious teachings, they learn easier." These suggest that psychosocial interventions should not only align with cultural and religious values but actively incorporate them as therapeutic resources. Such an approach may enhance children's engagement, strengthen community trust and increase the long-term effectiveness of support programs for displaced or returnee populations.

This study offers several novel insights that extend the existing literature. While the experiences of refugees during conflict have been extensively documented, the post-conflict period remains comparatively underexplored. In particular, although prior research has addressed adjustment challenges faced by refugees in host countries or by return migrants upon resettlement (Riiskjaer and Nielsson, 2008; Aghajafari et al., 2020), little attention has been given to IDPs who return home. This group encounters a uniquely complex context, as they are reintegrating into communities that now include both long-term residents and large numbers of refugees returning from host countries with diverse experiences. Furthermore, while the consequences of both forced and voluntary return have been widely discussed, this study highlights the often-overlooked phenomenon of children being subjected to family-driven forced return. Such cases complicate the assumption that post-conflict return is often positive, as children's place-based attachments and developmental needs may generate distinct psychological challenges. More broadly, the study contributes to a holistic understanding of how context-specific stressors, including socio-political environments, interact with children's mental health. It further highlights children's psychological resources from a context-sensitive perspective, such as the role of religious coping, while also underscoring the potential for PTG during this period. In terms of practical implications, the findings point to the importance of integrating individualized psychotherapy with psychosocial support, as the inconsistent outcomes of mental health interventions in conflict-affected settings may stem from insufficient integration of these approaches (Jordans et al., 2016). Moreover, sustaining a context-specific, culturally and religiously sensitive perspective in interventions requires strengthening the capacity of local NGOs, stakeholders and mental health professionals, and fostering greater collaboration among them.

This study has several strengths. First, while existing research has largely focused on refugee populations, this study directly examines internally displaced children living in an IDP camp, addressing a critical gap in the literature. Second, data collection shortly after the fall of the Assad regime provides a timely perspective on the post-conflict period and emerging challenges during political transition. Third, in-depth qualitative interviews enabled the collection of rich and nuanced data. The use of Syrian key informants with sustained professional engagement in the camp further strengthened the study by offering an insider perspective and contextual sensitivity. Several limitations should be noted. Direct interviews with children were not conducted due to ethical and safety considerations, limiting access to first-hand accounts. Although thematic saturation was reached, the relatively small and homogeneous sample may have constrained the diversity of perspectives. In addition, language barriers and logistical constraints required online interviews with interpreter support, which may have influenced data depth. Future research could address these limitations by ethically engaging children directly, including caregivers or families, expanding participant diversity and integrating quantitative measures alongside qualitative approaches to further strengthen evidence on children's mental health in post-conflict IDP settings.

### Conclusion

This study presents one of the first qualitative examinations of the post-conflict period in Syria, offering timely and valuable insights. It fills a critical gap in the literature by focusing on the return process of internally displaced children – a topic that has received far less attention than the return experiences of migrants. Despite certain logistical and language limitations, and the indirect assessment of children's needs through key informants, the findings are grounded in firsthand accounts from NGO staff who were themselves Syrian and actively working with children in the Atmeh region, thus providing insider perspectives rooted in local understanding. The study highlights that the mental health needs of internally displaced children in post-conflict Syria are deeply intertwined with their basic needs, sense of place and culturally and religiously informed psychological resources. These findings emphasize the importance of holistic, contextually grounded

interventions. From a broader perspective, the impact of such support may be further enhanced by efforts to facilitate reconstruction and improve socio-economic conditions. Addressing systemic restrictions that can complicate the delivery of aid and resources could be seen as a valuable step toward creating the stable environment children need to overcome past hardships. Future research should complement these qualitative findings with quantitative studies to better inform the design and implementation of psychological interventions in the field.

**Open peer review.** To view the open peer review materials for this article, please visit http://doi.org/10.1017/gmh.2026.10146.

**Supplementary material.** The supplementary material for this article can be found at http://doi.org/10.1017/gmh.2026.10146.

**Data availability statement.** The datasets generated and/or analyzed during the current study are not publicly available due to ethical and confidentiality reasons related to the sensitive status of IDP populations and NGO staff but are available from the corresponding author on reasonable request.

**Acknowledgments.** We sincerely thank all professionals who participated in the study for sharing their experiences. We would like to acknowledge the NGO by name in the acknowledgements section. We previously consulted the Editor-in-Chief and the Editorial Development Assistant regarding whether naming the NGO would be ethically appropriate. The NGO has provided consent to be acknowledged. We are currently awaiting editorial confirmation. If approved, we would like to include the following acknowledgement: 'We thank Yeryüzü Çocukları Derneği (a child-focused humanitarian NGO in Türkiye) for their support in facilitating participant recruitment.' We will follow the journal's guidance on this matter. The authors used OpenAI's ChatGPT (version GPT-5, accessed in 2025) to assist in improving the English language clarity and readability of the manuscript. The authors reviewed and verified all AI-assisted text to ensure accuracy and integrity. The authors used OpenAI's ChatGPT (version GPT-5, accessed in 2025) to assist in improving the English language clarity and readability of the manuscript. The authors reviewed and verified all AI-assisted text to ensure accuracy and integrity.

**Author contribution.** B.Ç.M.: conceptualization, data curation, formal analysis, project administration, writing – original draft. A.F.E.: data curation, formal analysis, writing – original draft. A.Ş.: methodology, supervision, writing – review & editing. All authors read and approved the final manuscript.

**Financial support.** This research received no specific grant from any funding agency in the public, commercial or not-for-profit sectors. The article processing charge will be covered under the open access agreement between Ibn Haldun University and Cambridge University Press.

**Competing interests.** The corresponding author (B.Ç.M.) previously volunteered with the NGO from which participants were recruited. However, this prior involvement did not influence the design, data collection, analysis or interpretation of the study. The authors declare no other competing interests.

**Ethics statements.** Ethical approval for this study was granted by the Ethics Committee of İbn Haldun University, Social and Human Sciences Scientific Research and Publication Ethics Board (Approval No: E-71395021-050.04-57,014). All participants provided written and verbal informed consent prior to participation. Consent included agreement for the interviews to be recorded and for anonymized quotations to be used in publications.

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
