## [Reviewer Report]

The study investigates the post-conflict period, specifically the phase following the regime’s fall which remains insufficiently examined in the child mental-health literature on Syria. However, there are important weaknesses in the methodology section. Despite being a qualitative study is justified, it provides no information about the inclusion and exclusion criteria, nor does it justify the limited number of participants. More detail is also needed regarding the questionnaire used to collect information.

Direct interaction with children would have offered deeper and more nuanced insights into their experiences. Additionally, incorporating quantitative information on the return-movement trends such as the scale, direction, and patterns of people moving back home would strengthen the study’s contextual grounding.

It is indeed true that the needs of children and the gaps in mental-health care are substantial. The return movement is likely to place additional pressure on both IDPs and those who had already been suffering under the regime. An important study in this field has provided valuable quantitative evidence on the existing literature concerning children’s health inside Syria, and it could be effectively used to enhance the discussion in this context (Post-crisis health reality and wellbeing of children within Syria: a scoping review of research from 2012 to 2024).

The study discusses the role of religious background effectively, highlighting how reliance on religion can support adaptation and help individuals overcome difficulties in such communities. However, it is also important to note that religion has at times played a negative role in fostering hate and escalating violence. Therefore, the discussion could be strengthened by recommending that religious teachings and practices be used to promote values such as compassion, empathy, patience, acceptance, and forgiveness, and by examining how these values can positively influence behavior.

Additionally, using art as a therapeutic method for children’s trauma is an excellent recommendation. One important reference that can be added to support this idea is “The Effectiveness of Self-Narrative Art Therapy in Reducing PTSD Symptoms among War-Affected Syrian Children.”

Finally, it would be valuable to emphasize the importance of facilitating reconstruction efforts and enabling the delivery of mental and physical support for children by easing restrictions and sanctions affecting Syrians everywhere. Improving the economy and living conditions for all Syrians would help reduce violence and support individuals in overcoming past hardships and losses.

---

## [Reviewer Report]

I thank the editor for giving me the opportunity to review this manuscript.The manuscript addresses a highly important and timely topic, and the qualitative insights provided by key informants working with internally displaced children in post-conflict Syria are valuable. The manuscript needs certain revisions before being considered for publication. You can find my general and specific comments below:

- I believe that the current title does not fully reflect the actual focus and contribution of the study. While the title suggests a direct examination of children’s mental health, the study is in fact based on key informant perspectives (NGO staff) and places substantial emphasis on contextual stressors, camp conditions, return-related challenges, coping resources, and service gaps, rather than on children’s mental health outcomes per se. therefore recommend revising the title to more accurately capture:

(a) the key informant–based nature of the study, and

(b) the broader ecological and contextual framing of children’s mental health in the post-conflict period. I just wanted to give an example: Mental Health Needs, Stressors, and Coping Resources of Internally Displaced Children in Post-Conflict Syria: A Qualitative Study with NGO Staff

The introduction is generally well written; however, it is somewhat indirect and overly expansive, which obscures the core problem addressed by the study. I recommend that the authors condense and sharpen the introduction by more explicitly addressing the following points:

• What is already known about internally displaced children living in camps, particularly in Syria

• What kinds of studies have been conducted to date (to situate it in larger literature), and what remains underexplored

• Whether there are existing NGO- or key informant–based qualitative studies in Syrian camps or comparable displacement contexts

• Why examining children’s mental health in camps during the post-conflict period is theoretically and practically important

Clarifying these points would help the authors more clearly articulate the literature gap and strengthen the unique contribution of the study.

The methodology section would benefit from further development and clarification. In particular, the authors should provide more detailed information on participant recruitment, including how participants were approached and recruited, through which channels, and what information was communicated to them regarding the purpose of the study.

In addition, the research questions could be presented more transparently, for instance by including them in an appendix, to enhance clarity and methodological transparency.

Regarding the qualitative analysis, the authors should clearly describe how the codes were developed, by whom, and through which analytic procedures. It would also be important to explain how many researchers were involved in the coding process and how consistency across coders was ensured.

Finally, the manuscript would benefit from a clearer discussion of methodological rigor, such as how interrater reliability and/or trustworthiness criteria (e.g., credibility, dependability, reflexivity) were established and maintained throughout the analysis

- Some of the themes appear to be insufficiently differentiated and conceptually overlapping. For instance, themes such as adverse circumstances in return areas and stressors in the camp seem to capture very similar underlying experiences and stressors, raising questions about their analytical distinctiveness.

The strengths of the manuscript should not be presented after the limitations section. The novelty of the study would be more appropriately articulated in the introduction, where the contribution of the work can be clearly positioned within the existing literature. While it is acceptable to acknowledge limitations and discuss strengths in relation to them within the same paragraph, the current organization would benefit from greater structural coherence.

---

## [Editor Report]

Thank you for submitting this timely and important manuscript. Reviewers appreciated the focus on the underexamined post-conflict period in Syria and the valuable qualitative insights from NGO staff working with internally displaced children. To strengthen the manuscript, they emphasized the need for clearer framing and methodological transparency, including sharpening the introduction to clearly articulate the literature gap and contribution and providing more detail on the methods. Reviewers also encouraged stronger contextual grounding and discussion, including engagement with relevant quantitative and qualitative literature, clearer differentiation of themes, and further description of the study strengths and limitations. I hope you will consider revising the manuscript according to these recommendations.

---

## [Editor Report]

Thank you for submitting your revised manuscript. We appreciate the thorough approach you took to responding to the comments from reviewers. The revised version has addressed the outstanding points raised during the review process.